# The Function of Mitochondrial Calcium Uniporter at the Whole-Cell and Single Mitochondrion Levels in WT, MICU1 KO, and MICU2 KO Cells

**DOI:** 10.3390/cells9061520

**Published:** 2020-06-22

**Authors:** Syed Islamuddin Shah, Ghanim Ullah

**Affiliations:** Department of Physics, University of South Florida, Tampa, FL 33647, USA; syedislamudd@mail.usf.edu

**Keywords:** mitochondrial Ca^2+^ uptake, mitochondrial Ca^2+^ uniporter, MICU1, MICU2, EMRE, Ca^2+^ overload

## Abstract

Mitochondrial Ca^2+^ ([Ca^2+^]_M_) uptake through its Ca^2+^ uniporter (MCU) is central to many cell functions such as bioenergetics, spatiotemporal organization of Ca^2+^ signals, and apoptosis. MCU activity is regulated by several intrinsic proteins including MICU1, MICU2, and EMRE. While significant details about the role of MICU1, MICU2, and EMRE in MCU function have emerged recently, a key challenge for the future experiments is to investigate how these regulatory proteins modulate mitochondrial Ca^2+^ influx through MCU in intact cells under pathophysiological conditions. This is further complicated by the fact that several variables affecting MCU function change dynamically as cell functions. To overcome this void, we develop a data-driven model that closely replicates the behavior of MCU under a wide range of cytosolic Ca^2+^ ([Ca^2+^]_C_), [Ca^2+^]_M_, and mitochondrial membrane potential values in WT, MICU1 knockout (KO), and MICU2 KO cells at the single mitochondrion and whole-cell levels. The model is extended to investigate how MICU1 or MICU2 KO affect mitochondrial function. Moreover, we show how Ca^2+^ buffering proteins, the separation between mitochondrion and Ca^2+^-releasing stores, and the duration of opening of Ca^2+^-releasing channels affect mitochondrial function under different conditions. Finally, we demonstrate an easy extension of the model to single channel function of MCU.

## 1. Introduction

Mitochondrial Ca^2+^ uptake plays a central role in cell metabolism, signaling, and survival [1,2,3]. Ca^2+^ entry into the matrix is mediated by the Ca^2+^ uniporter-channel complex with a high selectivity for Ca^2+^ and large carrying capacity [4,5,6,7,8]. This complex consists of mitochondrial Ca^2+^ uniporter (MCU), which constitutes the pore-forming subunit of the channel [5,9]. MCU interacts with several intrinsic proteins that regulate its activity. These include mitochondrial Ca^2+^ uptake protein 1 (MICU1) [10], MICU2 [11,12,13], MICU3 [14], MCU regulator 1 [15], and essential MCU regulator protein (EMRE) [16,17,18]. While MICU3 is largely restricted to brain, MICU1 and MICU2 are widely expressed and play major roles in mitochondrial Ca^2+^ uptake in most human cells [13,14]. Similarly, EMRE has been shown to play a key role in the regulation of MCU activity in the mitochondrial Ca^2+^ concentration ([Ca^2+^]_M_)-dependent manner [17,18,19]. 

Significant information about the role of regulatory proteins described above in the function of MCU has emerged over the last few years [10,11,12,13,14,15,18,19,20,21,22]. In particular, MICU1 has been shown to mediate the observed suppression of MCU activity in the low cytosolic Ca^2+^ concentration ([Ca^2+^]_C_) regime, called “gatekeeping” [12]. This gatekeeping of MCU activity is relieved when [Ca^2+^]_C_ reaches approximately 1.3 μM. Below this critical concentration, Ca^2+^ uptake through MCU is essentially negligible. MICU1 is also responsible for the observed highly cooperative activation of MCU activity where the gatekeeping is cooperatively relieved when [Ca^2+^]_C_ exceeds 1.3 μM [12] (Figure 1B). Similarly, MICU2 interacts with MICU1 to increase the [Ca^2+^]_C_ threshold for the gatekeeping and reduce the gain of cooperative activation of MCU activity [12] (Figure 1B). EMRE, on the other hand, plays a role in regulating MCU activity on the matrix side of the inner mitochondrial membrane (IMM) such that the uptake is minimum when [Ca^2+^]_M_ is approximately 400 nM [18,19]. Thus, MCU exhibits a biphasic behavior where the current through the channel increases as [Ca^2+^]_M_ drops below or rises above 400 nM (Figure 1C). Furthermore, MICU1 and MICU2 are required for the matrix Ca^2+^ regulation of MCU such that the inhibition at the intermediate [Ca^2+^]_M_ disappears when either one of the two proteins are knocked out [19].

Despite this wealth of information, many key issues about the activity of MCU and the role of different regulatory proteins in mitochondrial function remain unresolved and are beyond the scope of current experimental techniques. For instance, while experimental tools can be used to study the activity of MCU or how it is affected by a given regulatory protein at discrete fixed values of one or two variables, several variables affecting mitochondrial Ca^2+^ uptake in real cells vary continuously in mutually dependent manner. Furthermore, the observations about MCU activity are based on experiments performed at different spatiotemporal scales ranging from patch clamp electrophysiology of single Ca^2+^ uniporter channels and individual mitoplasts to the imaging of Ca^2+^ uptake at the cell culture level using fluorescence microscopy (e.g., see [6,11,12,19]). Linking these observations at different scales is key to understanding the role of Ca^2+^ in mitochondrial and cell function. For example, while significant data about mitochondria-dependent variables such as ATP and reactive oxygen species as a function of Ca^2+^ at the whole-cell or cell culture levels exists, mitochondrial Ca^2+^ uptake and function are mainly regulated by local Ca^2+^ concentration in a few tens of nanometers wide microdomain formed by the close apposition of plasma membrane or intracellular organelles with individual mitochondrion [23,24,25,26]. These and a range of other observations underscore the importance of incorporating the key observations about the role of regulatory proteins in the function of MCU in a comprehensive computational framework.

The importance of a computational model incorporating the role of regulatory proteins in the function of MCU is further highlighted by the role of these proteins in various pathologies. For example, the human loss-of-function mutations in MICU1 results in mitochondrial Ca^2+^ overload, impaired bioenergetics, and mitochondrial fragmentation [27]. These MICU1-induced defects in mitochondrial Ca^2+^ signaling and function are believed to result in early-onset neuromuscular weakness, impaired cognition, and extrapyramidal motor disorder [27,28,29]. A whole-body knockout of MICU1 has also been reported to result in perinatal lethality in mouse models [30,31]. In addition, the expression levels of EMRE were also altered in patients with MICU1 mutations compared to controls [27]. Recently, it has been shown that mutations in mitochondrial m-AAA proteases associated with spinocerebellar ataxia and hereditary spastic paraplegia inflict their cytotoxicity by affecting the biogenesis of EMRE. This leads to the accumulation of active MCU-EMRE channels lacking gatekeeping, which facilitates mitochondrial Ca^2+^ overload, opening of mitochondrial permeability transition pore, and neurodegeneration [32]. 

In this paper, we (1) develop a comprehensive data-driven model that reproduces the key observations about the function of MCU under a range of [Ca^2+^]_C_, [Ca^2+^]_M_, and mitochondrial membrane potential (MMP or Δψ) values in WT, MICU1 KO, and MICU2 KO cells; (2) combine the observations at the cell culture (whole-cell) and single mitoplasts levels into a single mathematical framework; (3) investigate how the Ca^2+^ signaling in the microdomain between the endoplasmic reticulum (ER) and mitochondrion compares in WT, MICU1 KO, and MICU2 KO mitoplasts by modeling the interaction between the ER-bound inositol 1,4,5-trisphosphate (IP_3_) receptor (IP_3_R) channel and MCU; (4) how does MICU1 or MICU2 KO affect mitochondrial function both in terms of Ca^2+^ uptake and ATP production; and (5) investigate the effect of Ca^2+^ buffers, width of the microdomain, and open duration of IP_3_R on mitochondrial Ca^2+^ uptake and ATP production in WT, MICU1 KO, and MICU2 KO cells.

## 2. Materials and Methods

The experimental data used in this paper is adapted from Riley et al. [12] and Vais et al. [19] with permission. These experiments were performed on HEK293 cells. We refer the interested reader to these two papers for details about experimental methods. 

### 2.1. Kinetic Model for MCU Function

Our kinetic model for MCU implements the scheme proposed in Payne et al. [12] and Vais et al. [19]. The model takes into account the explicit dependence of the channel’s open probability (*P_O_*) on [Ca^2+^]_C_, [Ca^2+^]_M_, and Δψ, combining a wide range of observations from patch clamp electrophysiology of individual mitoplasts and fluorescence microscopy of mitochondrial networks in cell cultures. The model assumes Ca^2+^ sensors on both sides of the IMM. On the cytosolic side, both MICU1 and MICU2 can bind up to two Ca^2+^ (both MICU1 and MICU2 have two EF hands each and each EF hand can bind one Ca^2+^) (Figure 1A). MICU1 suppresses the activity of MCU in the low [Ca^2+^]_C_ regime (0–1.3 μM), whereas MICU2 interacts with MICU1 to increase the [Ca^2+^]_C_ threshold for MCU activation and reduces the gain of cooperative activation of MCU activity. Thus, the channel is open when Ca^2+^ is bound to both EF hands of MICU1 and MICU2. On the matrix side, MCU activity is regulated by coupled inhibitory and activating Ca^2+^ sensors. At intermediate [Ca^2+^]_M_ (50–800 nM), Ca^2+^ binds to the inhibitory sensor that closes the channel (states *C_31_* and *C_41_* in the model shown in Figure 1A). At low [Ca^2+^]_M_, the matrix Ca^2+^-mediated gatekeeping is released (state *O_40_*). At high [Ca^2+^]_M_, Ca^2+^ binds to the activating senor, relieving the gatekeeping on the matrix side of the IMM. Our best fit criterion (BIC score) warrants the binding of three Ca^2+^ to the activating sensor in addition to the one Ca^2+^ that is bound in state *C_41_* to open the channel again (state *O_44_*). More recently, it was shown that the affinities of activating and inhibitory sensors on the matrix side are regulated by another Ca^2+^ sensor termed “flux sensor” [18]. The flux sensor is believed to represent a site on the matrix side of IMM that binds the Ca^2+^ as it enters through the channel. This sensor is sensitive to the Ca^2+^ buffering capacity of mitochondria, especially fast buffers that uptake Ca^2+^ immediately after entering the matrix. Although our study does not concern the role of mitochondrial Ca^2+^ buffers, our model can reproduce the observations about the role of “flux sensor” in mitochondrial Ca^2+^ uptake without changing the topology of the model (results not shown). We also remark that EMRE plays a key role in the matrix Ca^2+^-mediated regulation of MCU activity, and expressing mutant EMRE abolished this regulation. EMRE is also key to the strong coupling between the mechanisms regulating the channel activity on the matrix and the opposite side of IMM.

Note that, in our model, there are transitions that involve binding of more than one Ca^2+^. For example, the MCU has 0 and 2 Ca^2+^ bound to the sensors on the cytosolic side of IMM in states C_00_ and $C_{20}^{M1}$, respectively. However, the direct link between C_00_ and $C_{20}^{M1}$ does not necessarily mean that the channel binds 2 Ca^2+^ simultaneously. It rather means that the state with 1 Ca^2+^ bound have very low occupancy and is not required in the model. However, these transition states act as speed-bumps for the probability flux, and their effect can be incorporated in the mean transition times or transition rates. Similarly, for the connectivity of the model, we apply Occam’s razor and include the transitions that are warranted by Bayesian Information Criterion [33]. These issues have been discussed in detail elsewhere [34]. 

The above considerations lead to the kinetic scheme with six close (*C_XY_*) and two open (*O_XY_*) states (Figure 1A), where subscripts *X* and *Y* represent the number of Ca^2+^ bound to the domains on the cytosolic and matrix sides of the IMM, respectively. Relative to the reference unliganded close state *C_00_*, the occupancies (unnormalized probabilities) of the close *C_XY_* and open *O_XY_* states are proportional to ([Ca^2+^]_C_)^X^([Ca^2+^]_M_)^Y^ with occupancy parameters $K_{C_{XY}}$ and $K_{O_{XY}}$, respectively. The occupancy parameter of a state is the product of equilibrium association constants along any path connecting *C_00_* to that state ($K_{C_{00}}$ = 1) [34,35]. In other words, the product of all forward rates starting from *C_00_* to a given state *X* divided by the product of the rates from state *X* back to *C_00_* gives the occupancy of state *X*. This applies to all states and is independent of the path taken from *C_00_* to *X*. For example, the ratio of the transition rate from *C_00_* to $C_{20}^{M1}$ to the rate from $C_{20}^{M1}$ to *C_00_* gives the occupancy of $C_{20}^{M1}$. This will become clear further when we derive the transition rates for the single MCU channel in the Discussion section. We refer the interested reader to [34,35] for a more detailed discussion. Thus, the normalized occupancy of *C_XY_* and *O_XY_* are $K_{C_{\mathrm{XY}}}{\left({\left[{\mathrm{Ca}}^{2+}\right]}_{C}\right)}^{X}{\left({\left[{\mathrm{Ca}}^{2+}\right]}_{M}\right)}^{Y}/Z$ and $K_{o_{\mathrm{XY}}}{\left({\left[{\mathrm{Ca}}^{2+}\right]}_{C}\right)}^{X}{\left({\left[{\mathrm{Ca}}^{2+}\right]}_{M}\right)}^{Y}/Z,$ respectively, where *Z* is the total occupancy of all states and is given as
(1)$$\begin{array}{ll} Z=Z_{C}+Z_{O}= & \left\lceil \begin{array}{c} {1+K}_{C_{20}^{M1}}{\left({\left[{\mathrm{Ca}}^{2+}\right]}_{C}\right)}^{2}{+K}_{C_{20}^{M2}}{\left({\left[{\mathrm{Ca}}^{2+}\right]}_{C}\right)}^{2}{+K}_{C_{30}}{\left({\left[{\mathrm{Ca}}^{2+}\right]}_{C}\right)}^{3}\\ {+K}_{C_{31}}{\left({\left[{\mathrm{Ca}}^{2+}\right]}_{C}\right)}^{3}{\left[{\mathrm{Ca}}^{2+}\right]}_{M}{+K}_{C_{41}}{\left({\left[{\mathrm{Ca}}^{2+}\right]}_{C}\right)}^{4}{\left[{\mathrm{Ca}}^{2+}\right]}_{M} \end{array}\right\rceil \\ & +\left[K_{O_{40}}{{([\mathrm{Ca}}^{2+}{]}_{C})}^{4}{+K}_{O_{44}}{{([\mathrm{Ca}}^{2+}{]}_{C})}^{4}{{([\mathrm{Ca}}^{2+}{]}_{M})}^{4}\right] \end{array}$$
where *Z_C_* and *Z_O_* are the unnormalized occupancies of all close and all open states, respectively. The *P_O_* of the channel can be written as
(2)$$P_{O}\left({\left[{Ca}^{2+}\right]}_{C},{\left[{Ca}^{2+}\right]}_{M}\right)=\frac{Z_{O}}{Z_{C}}.$$
The Ca^2+^ uptake rate of MCU (*V_MCU_*) as a function of [Ca^2+^]_C_ and [Ca^2+^]_M_ is given by
(3)$$V_{\mathrm{MCU}}{=V}_{\mathrm{MCU},\max}\times P_{O}\left({\left[{\mathrm{Ca}}^{2+}\right]}_{C},{\left[{\mathrm{Ca}}^{2+}\right]}_{M}\right)$$
where $V_{\mathrm{MCU},\max}$ = 600 nM/s is the maximum uptake rate observed in the cell culture experiments [12]. We remark that the initial Ca^2+^ uptake rate for each experiment was determined by fitting an exponential function to [Ca^2+^]_C_ beginning from Ca^2+^ addition until a new steady-state is reached 300 s later to obtain parameters A (extent of uptake) and τ (time constant). The instantaneous rate of uptake at *t* = 0 was taken as equal to A/τ. In these experiments, only mitochondrial Ca^2+^ uptake through MCU was active, whereas all other Ca^2+^ pathways were either pharmacologically blocked or were inactive (see Ref. [12] for details).

### 2.2. Whole-Cell Ca^2+^ Model

In the cell culture experiments, HEK293 cells were treated with 0.004% digitonin at 50 s after the beginning of the experiment to permeabilize plasma membrane in intracellular-like medium lacking free Ca^2+^ buffers, equilibrating Ca^2+^ concentration in bath solution and cytoplasm. Ca^2+^ uptake into the ER and mitochondrial efflux were inhibited by applying 2 μM thapsigargin (Sarco/ER Ca^2+^ ATPase (SERCA) blocker) and 20 μM CGP37157 (Na^+^/Ca^2+^ exchange blocker) at 100 and 400 s, respectively. Similarly, no inositol 1,4,5-trisphosphate (IP_3_) was used in these experiments, implying no Ca^2+^ efflux from the ER through IP_3_ receptor (IP_3_R) channels. After [Ca^2+^]_C_ reached a steady-state at 700 s, MCU-mediated Ca^2+^ uptake was initiated by adding boluses of CaCl_2_ to achieve increases in [Ca^2+^]_C_ between 100 nM and 10 μM. All these considerations lead to the following rate equations for [Ca^2+^]_C_ and [Ca^2+^]_M_, respectively.
(4)$$\frac{{d[{\mathrm{Ca}}^{2+}]}_{C}}{\mathrm{dt}}=\left(V_{\mathrm{Stim}}{-V}_{\mathrm{MCU}}{+V}_{\mathrm{Leak}}\right),$$
(5)$$\frac{d{\left[{\mathrm{Ca}}^{2+}\right]}_{M}}{\mathrm{dt}}=f_{m}\times V_{\mathrm{MCU}}/\delta$$
where *V_Stim_* represents Ca^2+^ stimulus (bolus of CaCl_2_). The value of *V_Stim_* is set so that the peak [Ca^2+^]_C_ given by the model matches the observed value. Parameters *f_m_ =* 3 × 10^−4^ and *δ =* 0.067 are the Ca^2+^ buffering of mitochondria and the ratio of mitochondrial to cytosolic volume [36,37]. As we will see later, both of these parameters significantly affect free [Ca^2+^]_M_. We add *V_Leak_* to the rate equation for [Ca^2+^]_C_ to incorporate the effect of potential Ca^2+^ leak from the ER to the cytoplasm:(6)$$V_{\mathrm{Leak}}=V_{\mathrm{leak},\max}\left({\left[{\mathrm{Ca}}^{2+}\right]}_{\mathrm{ER}}-{\left[{\mathrm{Ca}}^{2+}\right]}_{C}\right),$$
where ${\left[{\mathrm{Ca}}^{2+}\right]}_{\mathrm{ER}}=200\;\mu M$ is Ca^2+^ concentration in the ER.

### 2.3. The Effect of MICU1 or MICU2 KO on Mitochondrial Function at the Whole-Cell Level

To assess how MICU1 or MICU2 KO affect mitochondrial function, we extend the model to incorporate the dynamics of mitochondrial NADH concentration ([NADH]_M_), mitochondrial ADP concentration ([ADP]_M_), Δψ, and cytosolic ADP concentration ([ADP]_C_). The rate equations, relevant fluxes (with the exception of *V_MCU_* and *V_NaCa_*), and parameters are adopted from Ref. [36] and are given in the Appendix A. Furthermore, Equation (5) is modified slightly to include Ca^2+^ efflux from the mitochondria through Na^+^/Ca^2+^ exchanger (*V_NaCa_*) [38].
(7)$$\frac{{d[{\mathrm{Ca}}^{2+}]}_{M}}{\mathrm{dt}}=f_{m}\times (V_{\mathrm{MCU}}-\;V_{\mathrm{NaCa}})/\delta ,$$
where
(8)$$V_{\mathrm{NaCa}}=V_{\mathrm{NaCa},\max}\frac{\left(\frac{{\left[{\mathrm{Ca}}^{2+}\right]}_{M}}{{\left[{\mathrm{Ca}}^{2+}\right]}_{C}}\right)\exp^{\left(\frac{\mathrm{bF}\left({\Delta \psi -\Delta \psi }_{0}\right)}{\mathrm{RT}}\right)}}{{\left(1+\frac{K_{\mathrm{Na}}}{{\left[\mathrm{Na}\right]}_{i}}\right)}^{3}\left(1+\frac{K_{\mathrm{Ca}}}{{{[\mathrm{Ca}}^{2+}]}_{M}}\right)},$$
where *b* = 0.5, *Δ**ψ_0_* = 91 mV, *K_Na_* = 9.4 × 10^3^ μM, and *K_Ca_* = 0.375 μM. [Na]_i_ = 11 × 10^3^ μM, *F*, *R*, and *T* = 310.16 ^o^K is the cytosolic Na^+^ concentration, Faraday’s constant, gas constant, and temperature, respectively. Note that *Δ**ψ_0_* = 91 mV does not represent the resting MMP. It was originally selected so that $V_{NaCa}$ is maximum when *Δ**ψ* = *Δ**ψ_0_*, i.e., under uncoupled conditions [38]. As we discuss later, *V_NaCa,max_* is estimated from the observed values of resting [Ca^2+^]_M_ in WT, MICU1, and MICU2 KO conditions. The full model used for mitochondrial function at the whole-cell level consists of Equations (7) (with Equations (3) and (8)) and Appendix A (Section “Bioenergetics model” in the Appendix A). Equation (3) is multiplied by Equation (9) given below to incorporate the voltage dependence of $V_{MCU}$.

## 3. Results

### 3.1. The Model Reproduces the Function of MCU both at Cell Culture and Single Mitoplast Levels

Model fits to the mitochondrial Ca^2+^ uptake rate as a function of [Ca^2+^]_C_ observed in cell cultures using fluorescence microscopy on WT (spheres, solid line), MICU1 KO (squares, dashed line), and MICU2 KO (diamonds, dashed-dotted line) cells are shown in Figure 1B. The values of occupancy parameters for the three conditions are given in Appendix A. The model closely reproduces the general features of the MCU activity including the suppressed activity in the low [Ca^2+^]_C_ regime (0–1.3 μM), the cooperative opening once the gatekeeping is relieved, relief of gatekeeping and reduced cooperative activation in MICU1 KO cells, and reduced threshold for gatekeeping and increased gain of cooperative activation in MICU2 KO cells. We remark that our fitting criterion searches for the parameters that result in the best fit to all observations simultaneously. For example, in case of MICU1 KO experiments, we do not restrict our search to states that are associated with MICU1 only. In other words, we make no a priori assumption about the independent binding of Ca^2+^ to the MICU1 EF hands. This is motivated by the complex structure of MCU [5,8,9,39] and the understanding that “the protein is not a rigid system in which a ligand moves in a fixed potential. Instead, there is a strong mutual interaction between ligand and protein” [40] that affects several aspects of the channel gating. This approach leads to an interesting observation in our model: MICU1 (or MICU2) KO not only affects the probability of states associated with MICU1 but also change the occupancy of other binding sites (Appendix A). 

Notice that the model uses the values of both [Ca^2+^]_C_ and [Ca^2+^]_M_ for the fit. However, the experiments on cell cultures report the uptake rate as a function of [Ca^2+^]_C_, whereas the values of [Ca^2+^]_M_ at the [Ca^2+^]_C_ used are not known. In one set of experiments, Vias et al. [19] measured [Ca^2+^]_C_ and [Ca^2+^]_M_ simultaneously (without measuring MCU activity) (see Figure 3J,K in Ref. [19]). We use the values from those experiments to derive a rough estimate of [Ca^2+^]_M_ as a function of [Ca^2+^]_C_ in MICU2 KO cells. That is, we plot [Ca^2+^]_M_ versus [Ca^2+^]_C_ from these experiments and interpolate this relationship to other [Ca^2+^]_C_ values. These estimates are crude because the peak [Ca^2+^]_C_ value used in those experiments was less than 1 μM, whereas the values used in Figure 1B go beyond 8 μM. We believe that this information gap could be causing the discrepancy between the model and observations in the MICU2 KO cells (Figure 1B). We noticed that another set of parameters give a closer fit to the uptake rate observed in the MICU2 KO cell culture, but it deteriorates the fit to the MCU current (I_MCU_) observed at the single mitoplast level [19] (Appendix A). Since both [Ca^2+^]_C_ and [Ca^2+^]_M_ are known in the patch clamp experiments, we choose the first set of parameters (Appendix A) over the second for the rest of this paper. 

Payne et al. [12] suggested that the state with Ca^2+^ bound to both EF hands of MICU1 and one EF hand of MICU2 is an open state. To test this hypothesis, we change the state *C_30_* in the model to an open state (*O_30_*). However, we found that such change is not supported as the model predicts higher Ca^2+^ uptake rate in WT cells (Appendix A). Furthermore, the overall fit to the observed uptake rate in MICU1 KO cells significantly deteriorates (Appendix A). We therefore leave this state as a close state from now on. 

In our formalism, the transition from whole-cell model (based on cell culture experiments) to the single mitoplast model is straightforward. The only parameter that changes while fitting to the single mitoplast data is that $V_{MCU,max}$ in Equation (3) is replaced by the maximum value of MCU current density (I_MCU_/C_m_) observed in patch clamp experiments, which is equal to 206, 237, and 217 pA/pF in case of WT, MICU1 KO, and MICU2 KO mitoplasts, respectively. The model fits a range of observations about individual mitoplasts. The theoretical curves mirror the inverted bell-shaped behavior of I_MCU_/C_m_ as a function of [Ca^2+^]_M_ and the lack of inhibition in the MICU1 KO and MICU2 KO mitoplasts (Figure 1C). We also look at the maximum value of I_MCU_/C_m_ at different [Ca^2+^]_C_ and fixed [Ca^2+^]_M_ = 0.4 μM. Apart from the apparent mismatch between theory and experiment in case of MICU1 KO mitoplasts (Figure 1D, triangles and dashed line), the model exhibits the same general trend in WT and MICU2 KO mitoplasts (Figure 1D). The discrepancy in case of MICU1 KO could be due to the fact that the model represents MICU1 KO mitoplasts, whereas the experimental data are from MICU1 knockdown mitoplasts and is shown only for a crude comparison due to the lack of such data on MICU1 KO mitoplasts. Indeed, the overall theoretical trend is consistent with the observations in Figure 1C (squares and dashed line) where the current through MCU in MICU1 KO mitoplasts is larger than both the WT and MICU2 KO mitoplasts. The model reproduces MCU current density at different [Ca^2+^]_C_ and fixed [Ca^2+^]_M_~0 μM (Figure 1E, spheres and solid line). While we do not have experimental data on MICU1 KO and MICU2 KO mitoplasts, model predictions for these two cases are shown in Figure 1E by the dashed and dashed-dotted lines, respectively. 

### 3.2. The Relief from Mitochondrial Ca^2+^-Mediated Gatekeeping at Low [Ca^2+^]_M_ Decreases as We Decrease [Ca^2+^]_C_

We saw in the previous section that the MCU current density exhibits an inverted bell-shaped behavior as we change [Ca^2+^]_M_ with a minimum (a maximum suppression of 75%) at 0.4 μM at [Ca^2+^]_C_ = 1 mM in WT mitoplasts (Figure 1C, spheres and solid line). This matrix Ca^2+^-mediated gatekeeping is relieved when [Ca^2+^]_M_ decreases or increases from 0.4 μM. As the model enables us to estimate the current density as we vary [Ca^2+^]_C_ and [Ca^2+^]_M_ simultaneously in a continuous manner, we next tested this inverted bell-shaped behavior at varying [Ca^2+^]_C_. We found that, while the gatekeeping remains intact, the recovery from it (on the low [Ca^2+^]_M_ side) shifts to even lower [Ca^2+^]_M_ values and eventually disappears as we decrease [Ca^2+^]_C_ (Figure 2A). As is clear from Figure 2B, this behavior is also supported by observations where we show the ratio of MCU current density observed in WT mitoplasts at [Ca^2+^]_M_ = 0 μM to that observed at [Ca^2+^]_M_ = 0.4 μM at different [Ca^2+^]_C_ values. The ratio decreases significantly as we decrease [Ca^2+^]_C_, eventually reaching 1 at [Ca^2+^]_C_ = 0 μM. On the high [Ca^2+^]_M_ side, the recovery from gatekeeping shifts to even higher [Ca^2+^]_M_ values as we decrease [Ca^2+^]_C_ (Figure 2A). In other words, the range of [Ca^2+^]_M_ values where gatekeeping is effective gets narrower as [Ca^2+^]_C_ increases. This, together with the cooperative opening of MCU once the gatekeeping on the cytosolic side is relieved, could potentially lead to a vicious cycle, leaving mitochondria increasingly vulnerable to Ca^2+^ overload and the opening of mitochondrial permeability transition pore [41] as [Ca^2+^]_C_ increases beyond a certain limit. 

### 3.3. Membrane Potential Dependence of MCU Current Density

Membrane potential (Δψ) is another key variable influencing current through MCU by providing a driving force for Ca^2+^ influx. Figure 2C shows that the inward current density through MCU decreases as we make mitochondrial membrane more depolarized. While the amplitude of I_MCU_/C_m_ changes as we change [Ca^2+^]_C_, [Ca^2+^]_M_, and knockout MICU1 or MICU2, normalizing all traces with respect to their respective peak values shows that they follow the same functional form with respect to Δψ. Thus, we use the following function to represent the Δψ dependence of Ca^2+^ flux through MCU: (9)$$f(\Delta \psi )=A{\left(-\Delta \psi \right)}^{b}$$
where A = 1.62349 and b = 2.69 × 10^−4^. Fit to I_MCU_/C_m_ given by the model is shown by the thick solid line in Figure 2D. Multiplying Equations (3) and (9) leads to a complete model for MCU Ca^2+^ uptake rate as a function of [Ca^2+^]_C_, [Ca^2+^]_M_, and Δψ. In Figure 2E, we show I_MCU_/C_m_ predicted by the model as a function of [Ca^2+^]_M_ and Δψ at [Ca^2+^]_C_ = 100 μM. We choose this value for [Ca^2+^]_C_ because of its physiological relevance, as Ca^2+^ concentration in the microdomain formed by the close apposition of mitochondrion with the ER or plasma membrane reaches similar values when the Ca^2+^ channels on these membranes are open [23,42,43]. 

The interplay between [Ca^2+^]_C_, [Ca^2+^]_M_, and Δψ constitutes a robust signaling mechanism, controlling mitochondrial function. For example, raising [Ca^2+^]_C_ by a few μM increases [Ca^2+^]_M_ by a few tens of μM [44,45]. Such concentrations would cause the gatekeeping on both sides of the IMM to relieve, resulting in a maximum flux through MCU [12,19]. Furthermore, normalized TMRE fluorescence representing membrane potential (Δψ) increases by more than 0.1 on a 0.5 to 1 scale (where 1 represents the complete dissipation of the potential difference across IMM) when [Ca^2+^]_C_ is raised by 5 μM in patch clamp experiments on WT mitoplasts (see Figure S3 in Ref. [12]). The drop in Δψ would cause a decrease in Ca^2+^ uptake by mitochondria, giving another crucial control to these key variables to regulate mitochondrial function. We remark that all analysis in Figure 2 is applicable to MCU function at the cell culture level except that the peak I_MCU_/C_m_ (206 pA/pF) gets replaced by the maximum uptake rate observed at the whole-cell level (600 nM/s). 

### 3.4. Model Fits to the Whole-Cell Ca^2+^ Signals

Before making predictions, we show that the model closely fits the observed whole-cell cytosolic Ca^2+^ dynamics when WT, MICU1 KO, and MICU2 KO cells are challenged with an acute increase of different amounts of [Ca^2+^]_C_ in the cell culture experiments (Figure 3). At very high [Ca^2+^]_C_ (>7 μM), the higher *V_MCU_* causes [Ca^2+^]_C_ in MICU2 KO to drop significantly faster than both WT and MICU1 KO cells (Figure 3A). As [Ca^2+^]_C_ drops to lower values, *V_MCU_* in MICU1 KO cells remains elevated longer than that in WT and MICU2 KO cells, bringing [Ca^2+^]_C_ in MICU1 KO cells below that in WT and MICU2 KO cells. The trajectories at lower [Ca^2+^]_C_ bolus also follows a similar trend (Figure 3B). 

The peak [Ca^2+^]_M_ (peak free Ca^2+^ in the mitochondria) given by the model is an order of magnitude lower than that observed experimentally (Figure 3C,D). For example, raising [Ca^2+^]_C_ close to 800 nM in WT cells increases [Ca^2+^]_M_ to more than 1 μM (Figure 3J,K in Ref. [12]). However, the Ca^2+^ buffering capacity of mitochondria (*f_m_*) and the relative volume of mitochondria with respect to cytosol (*δ*) both play key roles in the free [Ca^2+^]_M_. We repeat the simulation in Figure 3A for WT cells using Ca^2+^ bolus of different strengths and record the peak [Ca^2+^]_C_ during the simulation and [Ca^2+^]_M_ 300 s after the Ca^2+^ bolus is added to the cytoplasm. As shown in Figure 3E,F, the peak free mitochondrial Ca^2+^ increases dramatically as we increase *f_m_* (decreasing mitochondrial Ca^2+^ buffering capacity) or decrease *δ* (decreasing the relative mitochondrial volume). Using the simultaneously measured values of [Ca^2+^]_C_ and [Ca^2+^]_M_ from Ref. [12] as a reference, we found that *f_m_* ~ 0.09 results in a close correspondence between [Ca^2+^]_C_ and [Ca^2+^]_M_ when *δ =* 0.067 is considered. The *f_m_* value giving the correct correspondence between model and observations would decrease (higher mitochondrial Ca^2+^ buffering capacity) as we decrease *δ*. 

### 3.5. Mitochondrial Bioenergetics in WT, MICU1 KO, and MICU2 KO Cells

In simulations shown in Figure 3, Δψ is kept fixed at −160 mV. However, Δψ changes as mitochondria buffers Ca^2+^. We therefore extend our model for mitochondrial Ca^2+^ uptake at the whole-cell level to incorporate the dynamic behavior of Δψ and how mitochondrial NADH ([NADH]_M_) and ATP ([ATP]_M_) change as mitochondria buffer Ca^2+^ in WT, MICU1 KO, and MICU2 KO cells. A key parameter missing in our whole-cell Ca^2+^ model is the maximum flux through Na^+^/Ca^2+^ exchanger (*V_NaCa,max_* in Equaiton (8)). Although we do not have direct access to this parameter in our experimental results, we use the observed resting value of [Ca^2+^]_M_ to estimate *V_NaCa,max_.* Knowing that the Ca^2+^ fluxes through MCU and Na^+^/Ca^2+^ exchanger balance each other out in resting conditions [27], and the MCU flux is constrained by the observations discussed above, we vary *V_NaCa,max_* until [Ca^2+^]_M_ given by the model in resting state matches the observed value (Appendix A). The values given by the fit in WT, MICU1 KO, and MICU2 KO conditions at the whole-cell level are given in the legends of Appendix A. In line with the observations where the deregulated mitochondrial Ca^2+^ uptake due to the loss of MICU1 function is shown to initiate a futile cycle whereby continuous Ca^2+^ influx is balanced by efflux through Na^+^/Ca^2+^ exchange [27], our model predicts higher *V_NaCa,max_* in MICU1 KO cells (2.144 nM/s) when compared to WT cells (0.958 nM/s). 

After reaching steady state, we apply a square pulse of Ca^2+^ to the cytoplasm, raising [Ca^2+^]_C_ from its resting value of 0.1 μM to 1.5 μM to see how key mitochondrial variables respond to the [Ca^2+^]_C_ rise in the three conditions. Typical time traces of [Ca^2+^]_M_, Δψ, [NADH]_M_, and [ATP]_M_ are shown in Appendix A. In resting condition, MICU1 KO cell has higher [Ca^2+^]_M_ (Appendix A) and slightly hyperpolarized IMM (Appendix A) when compared with WT and MICU2 KO cells, which mainly results from the higher [NADH]_M_ (Appendix A) that also leads to higher [ATP]_M_ in resting conditions (Appendix A). The small depolarization in IMM as soon as [Ca^2+^]_C_ is increased (note the very small increase in Δψ before the relatively large decrease) results from the Ca^2+^ and Na^+^ influx through MCU and Na^+^/Ca^2+^ exchanger, respectively. The small depolarization is followed by a relatively large hyperpolarization (although still very small in absolute terms), mainly due to the increase in [NADH]_M_ that in turn results from the stimulation of the pyruvate dehydrogenase reaction. While the values of [NADH]_M_, Δψ (more hyperpolarized), and [ATP]_M_ in resting conditions are larger, the relative change in all these variables in response to the Ca^2+^ bolus is smaller in MICU1 KO cells as compared to both WT and MICU2 KO cells. The relative smaller increase in [ATP]_M_ in response to [Ca^2+^]_C_ rise is consistent with the observations by Bhosale et al. [27] showing that the futile Ca^2+^ cycle due to uninhibited MCU influx balanced by larger efflux through Na^+^/Ca^2+^ exchanger leads to a lower ATP production in cells with MICU1 mutations. 

### 3.6. Ca^2+^ Uptake and Bioenergetics at the Single Mitoplast Level

There are two main discrepancies between Δψ given by the model at the whole-cell level and that observed experimentally in the cell culture experiments (see Figure S3 in Ref. [12]). First, the observed IMM potential in MICU1 KO cells is more depolarized than the WT and MICU2 KO cells. Second, the observed depolarization of IMM after the application of Ca^2+^ bolus is significantly larger when compared to the model predictions where the depolarization due to Ca^2+^ influx is almost unnoticeable. This is mainly due to the fact that the estimated maximum flux through MCU at the whole-cell level is small as it represents the average uptake of the entire mitochondrial network in the cell culture. That is, it represents the rate at which the average extra-mitochondrial Ca^2+^ in the cell culture decreases due to the net uptake by the entire mitochondrial network. Similarly, [Ca^2+^]_C_ and [Ca^2+^]_M_ also represent the average concentrations and ignore the large concentration gradients that could occur in the microdomain formed by the close apposition of a mitochondrion and the ER. In reality, a mitochondrion experiences Ca^2+^ on the cytosolic side that could be significantly higher than the bulk [Ca^2+^]_C_. Furthermore, as shown below, the maximum flux through MCU estimated from the patch-clamp experiments on single mitoplasts comes out to be significantly larger. We would like to point out that the experiments in [19] investigated the function of MCU at the single mitoplast level, not the bioenergetics of the mitochondrion. Thus, we model bioenergetics at the single mitochondrion level as if it is residing inside a living cell, but the MCU functions the same way as it would in the mitoplast. 

The maximum value of MCU current density (I_MCU_/C_m_) observed in the patch-clamp experiments is equal to 206, 237, and 217 pA/pF in the case of WT, MICU1 KO, and MICU2 KO mitoplasts respectively at *Δ**ψ* = −160 mV [19]. These values can be used to estimate the maximum flux through MCU (*V_MCU,max_* in Equation (3)) as follows:(10)$$V_{\mathrm{MCU},\max}=\frac{I_{\mathrm{MCU}}}{{2\times F\times \mathrm{Vol}}_{m}}$$
where *Vol_m_* is the volume of the mitochondrion. We consider a spherical mitochondrion of radius 0.5 μm. Vais et al. [19] observed the mitoplast capacitance (*C_m_*) to be in the range 0.2–1 pF. We use *C_m_* = 0.2 pF, giving I_MCU_ = 41.2, 47.4, and 43.4 pA for WT, MICU1 KO, and MICU2 KO cells. 

In intact cells, mitochondrial Ca^2+^ uptake at the single mitochondrion level also depends on the spatial distance between the ER and mitochondrion. Ca^2+^ is released from the ER through IP_3_R that diffuses away from the channel in the microdomain (Appendix A). We consider a single IP_3_R, allowing it to open for 100 ms, and model Ca^2+^ concentration in the microdomain ([*Ca^2+^*]*_mic_*) as a function of distance from the channel (*r*) and current through IP_3_R (*I_IP3R_*) using the following equation [46,47]:(11)$${\left[{\mathrm{Ca}}^{2+}\right]}_{\mathrm{mic}}=\frac{I_{\mathrm{IP}3R}}{{4\pi \mathrm{rD}}_{\mathrm{Ca}}}\exp^{\left(-\frac{r}{\lambda }\right)}+{\left[{\mathrm{Ca}}^{2+}\right]}_{\mathrm{rest}},$$
where *I_IP3R_ =* 0.05 pA [48], [*Ca^2+^*]*_rest_* = 100 nM, and *D_Ca_* = 223 μm^2^/s [49] is the current flowing through an open IP_3_R, [Ca^2+^]_C_ at rest, and diffusion coefficient of Ca^2+^. *λ* is given by
(12)$$\lambda =\sqrt{\frac{D_{\mathrm{Ca}}}{k_{\mathrm{on}}B_{T}\;K_{d}/(K_{d}{+\mathrm{Ca}}_{\mathrm{rest}})}}$$

*B_T_* is the total concentration of Ca^2+^ binding protein with *K_d_* = 2.0 μM [50]. Appendix A shows examples of Ca^2+^ concentration in the microdomain given by Equation (11) as a function of distance from the IP_3_R at different buffer concentrations. 

Thus, the full model used for mitochondrial function at the single mitoplast level consists of Equations (7) (with Equations (3), (8) and (10)), (11), Appendix A. Equation (3) is multiplied by Equation (9) to incorporate the voltage dependence of $V_{MCU}$. We remove the parameter *δ* in Equation (7) for simulating the single mitoplast because $V_{MCU}$ in the whole-cell experiments was estimated based on the dynamics of extra-mitochondrial Ca^2+^ concentration, and we needed to factor-in the relative size of mitochondria with respect to the cytoplasm. In case of single mitoplast experiments, $V_{MCU}$ is estimated directly from the current flowing into the mitoplast through MCU and such factoring is not required. 

Like the whole-cell model, the maximum flux through Na^+^/Ca^2+^ exchanger at the single mitoplast level is determined by fitting the resting value of [Ca^2+^]_M_ given by the model to the observed values (Appendix A). The values for *V_NaCa,max_* in WT, MICU1 KO, and MICU2 KO are given in the legends of Appendix A. The fit results in a significantly large *V_NaCa,max_* in MICU1 KO mitoplasts as compared to WT mitoplasts. This is in line with recent observations from whole-genome data from Alzheimer’s disease patients where significant upregulation of genes encoding Na^+^/Ca^2+^ exchangers is observed when compared to age-matched healthy individuals (see also the Discussion section below) [51]. After reaching steady state during the first 100 s of simulations, the IP_3_R channel is allowed to open for 100 ms. Typical time-traces of [Ca^2+^]_M_, Δψ, [NADH]_M_, and [ATP]_M_ are shown in Appendix A, where we consider a 24 nm wide microdomain [52]. A comparison between mitochondrial dynamics at the whole-cell (Appendix A) and single mitoplast (Appendix A) levels reveals four key differences. First, instead of hyperpolarization as in the case of whole-cell model, the Ca^2+^ uptake leads to the depolarization of IMM at the single mitoplast level (Appendix A)—in line with the observations in Ref. [12]. Second, in the resting conditions, IMM in the MICU1 KO cells is more depolarized than WT and MICU2 KO cells, which is also consistent with observations [12]. Third, [ATP]_M_ under resting conditions is lowest in MICU1 KO cells followed by MICU2 KO and WT cells (Appendix A). Finally, the mitochondrial Ca^2+^ in MICU1 KO cells results in [ATP]_M_ depletion due to the large depolarization that overshadows the rise in [NADH]_M_ (Appendix A). Furthermore, the Ca^2+^ influx in MICU1 KO mitoplasts results in significantly large depolarization of IMM as compared to the other two conditions. As we discuss in the next section, this is also in line with recent observations [18]. 

Csordás et al. found that the average spatial width of the microdomain is ~24 nm in RBL-2H3 cells [52]. However, significant variability exists in the microdomain width where its value can vary from a few nanometers to several hundred nanometers [53,54]. The resting and maximum values of [Ca^2+^]_M_, Δψ, [NADH]_M_, and [ATP]_M_ from the model as we increase the width of the microdomain in WT, MICU1 KO, and MICU2 KO cells are shown in Figure 4. While the resting values (dotted lines) remain nearly unchanged, the maximum values (solid lines) of [Ca^2+^]_M_ (A), Δψ (B), [NADH]_M_ (C), and [ATP]_M_ (D) in WT and MICU2 KO cells decrease significantly as we increase the width of the microdomain. This occurs because the mitochondrion is exposed to lower extra-mitochondrial Ca^2+^ concentration as we increase the width of the microdomain (longer distance between the IP_3_R and MCU). However, the decrease in the maximum values of the four variables is not as dramatic in the MICU1 KO mitoplast, even if we consider the microdomain to be a few hundred nanometers wide. Furthermore, the change in [ATP]_M_ in MICU1 KO cells is significantly lower in addition to the fact that [ATP]_M_ decreases in response to [Ca^2+^]_C_ rise (note that the “maximum” value of [ATP]_M_ after the channel opens is smaller than the resting value). 

We repeat the simulations in Appendix A varying the duration for which the IP_3_R channel is open and analyze the changes in the mitochondrial variables as [Ca^2+^]_mic_ rises. As is clear from Figure 5, the change in [Ca^2+^]_M_ (Figure 5A), Δψ (Figure 5B), [NADH]_M_ (Figure 5C), and [ATP]_M_ (Figure 5D) with respect to their resting values gets larger as we increase the open duration of the channel from 10 ms to 1 s incrementally. The larger mitochondrial Ca^2+^ uptake in MICU1 KO cells results in larger depolarization of IMM than MICU2 KO and WT cells (Figure 5B). Interestingly, the relative change in [NADH]_M_ in MICU1 KO mitoplasts is smaller than both WT and MICU2 KO cells despite the larger mitochondrial Ca^2+^ influx (Figure 5C).

To investigate this counterintuitive result further, we look at the three fluxes that control [NADH]_M_ in the model (Appendix A). As expected, the pyruvate dehydrogenase-catalyzed reaction (Appendix A) leading to NADH production increases with [Ca^2+^]_M_ (Appendix A) [55,56]. However, the resting [Ca^2+^]_M_ in MICU1 KO cells (~0.258 μM) is in the range where the reaction is almost saturated already and any further increase in [Ca^2+^]_M_ would lead to little change in this reaction. The second contribution to [NADH]_M_ in the model comes from the malate-aspartate shuttle (MAS) [57,58]. The model takes into account the activation of two mammalian carriers aralar and citrin involved in MAS by moderate cytosolic Ca^2+^ increases (Appendix A). By activating the Krebs cycle, the rise in [Ca^2+^]_M_ would lead to a decrease in the amount of α-ketoglutarate, a key-metabolite of the MAS [57,58]. This limiting step is taken into account in the model [36] and the flux due to MAS shuttle decreases with [Ca^2+^]_M_ and is almost at the minimum value in MICU1 KO cells (Appendix A) where the resting [Ca^2+^]_M_ ~ 0.26 μM (Appendix A). Finally, in the model, the oxidation of NADH in the Electron Transport Chain and the coupled extrusion of protons from the mitochondria are incorporated in a single equation (Appendix A) [36,38,59,60,61,62,63]. This flux declines for more hyperpolarized IMM, since it is difficult to pump protons against a large potential gradient and vice versa (Appendix A). The relatively more depolarized IMM in resting state in MICU1 KO cells results in a larger value of this flux that decreases [NADH]_M_ further. 

The large increase in [NADH]_M_ due to the rise in [Ca^2+^]_M_ in WT and MICU2 KO mitoplasts results in a significant increase in [ATP]_M_ which increases further as the open duration of the channel increases (Figure 5D). Contrary to WT and MICU2 KO, the large depolarization of the IMM coupled with a relatively small increase in [NADH]_M_ in MICU1 KO cells leads to a drop in [ATP]_M_. The larger resting value of [NADH]_M_ and smaller increase in response to cytosolic Ca^2+^ in MICU1 KO cells is consistent with observations where a larger concentration of NADH in resting state and smaller change in response to Histamine-induced rise in intracellular Ca^2+^ was observed in primary fibroblasts with MICU1 mutations in individuals with a disease phenotype characterized by proximal myopathy, learning difficulties, and a progressive extrapyramidal movement disorder as compared to those from age-matched healthy individuals [28]. The difference between the changes in [NADH]_M_ and [ATP]_M_ in MICU1 KO mitoplasts with respect to WT cells grow as we increase the open duration of IP_3_R, indicating that, in the face of prolonged [Ca^2+^]_C_ increases, MICU1 KO cells would be more vulnerable to damage due to the lack of ATP to perform necessary cell functions. 

To test whether adding Ca^2+^ buffer to the cytoplasm would restore mitochondrial function in MICU1 KO cells, we vary the buffer concentration from 0 to 1000 μM. Like the results shown in Figure 4, the peak value of [Ca^2+^]_M_ decreases significantly in WT and MICU2 KO cells as we increase the buffer concentration (Appendix A). However, both the maximum and resting values of [Ca^2+^]_M_ in MICU1 KO cells remain nearly constant. The values of other variables follow the same trend and are not shown. We remark that the kinetics of the buffer used in these simulations are relatively slow. A high concentration of a buffer with significantly faster kinetics that could immediately uptake most of the Ca^2+^ released by IP_3_R before it reaches the mitochondrion would decrease the peak value of [Ca^2+^]_M_ (Appendix A). Nevertheless, the resting values of [Ca^2+^]_M_, [ATP]_M_, and other variables remain unchanged from those shown in Appendix A (dashed lines in Appendix A).

## 4. Discussion

While the sigmoidal dependence of mitochondrial Ca^2+^ uptake on cytosolic Ca^2+^ was observed 60 years ago [64,65], details about how MICU1, MICU2, and EMRE define this relationship by regulating the gating of MCU began to emerge recently [8,66]. The functional importance of these regulatory proteins is further highlighted by their crucial role in various pathologies [27,28,29,30,31,32]. A key challenge for future experiments is to investigate how these regulatory proteins modulate mitochondrial Ca^2+^ influx through MCU in intact cells under physiological and pathological conditions, and how the interaction between MCU and regulatory proteins affects the bioenergetics of mitochondria in particular and cell signaling in general. This task is further complicated by the fact that mitochondrial Ca^2+^ uptake depends on the mitochondrial organization in the cell, the size of individual mitochondrion, the spatial positioning of a single mitochondrion with respect to other Ca^2+^-releasing organelles and channels, and various Ca^2+^ buffering proteins inside and outside the mitochondria. To make matters worse, mitochondria are in a constant state of transporting, fission, fusion, and fragmentation as the local Ca^2+^ gradients and metabolic demands change dynamically during cell functioning. Existing experimental techniques are not equipped to address the majority of these and other related issues. Data-driven computational models closely reproducing key observations about MCU function and its dependence on various variables and modulators offer a viable alternative. 

Several models for mitochondrial Ca^2+^ signaling have been developed over the past (for example, see [36,38,47,59,61,62,63,67,68,69,70]). However, they are mostly based on Ca^2+^ uptake in permeabilized cell cultures or isolated mitochondria suspended in aqueous solutions. These models do not properly incorporate the fine details of MCU’s gating kinetics, which could lead to erroneous conclusions [70]. The model in [71] links the biophysical properties of MCU activity to the mitochondrial Ca^2+^ uptake at the whole-cell level using Michaelis–Menten type relationship, but it does not incorporate the regulation of MCU by MICU1, MICU2, or EMRE. Similarly, the model in [72] investigates the mitochondrial Ca^2+^ dynamics in cells and isolated mitochondria in suspensions, but does not incorporate the effects of these proteins. Our model not only reproduces many key observations about MCU function at the single mitochondrion and whole-cell levels but also incorporates the role of mitochondrial membrane potential, MICU1, MICU2, and EMRE at both scales. Our analysis reaffirms that the three regulatory proteins together with MMP, the spatial positioning of the mitochondria with respect to the Ca^2+^ releasing channels or organelles, and the spatiotemporal dynamics of Ca^2+^ concentration and buffering proteins make for a robust and sophisticated signaling machinery for regulating mitochondrial metabolic function. 

Our model provides an excellent platform for investigating several of the issues outlined above. As an example, we investigate the interaction of a single IP_3_R with MCU in a microdomain and show that the separation between the two channels, the concentration of Ca^2+^ buffer in the microdomain, the open duration of IP_3_R, and the regulatory proteins all play key roles in mitochondrial metabolic function. Interestingly, our microdomain model shows that, while Ca^2+^ release through IP_3_R results in an increased ATP in WT and MICU2 KO mitoplasts, the exaggerated Ca^2+^ uptake in MICU1 KO mitochondrion results in a significant drop in ATP. The rise in ATP and [Ca^2+^]_M_ in response to the opening of IP_3_R in WT and MICU2 KO conditions decreases as we increase the width of the microdomain within the experimentally observed range or increase the concentration of Ca^2+^ buffer. Both of these parameters do not affect the metabolism of MICU1 KO mitochondria. A high concentration of fast buffers is needed to prevent the mitochondrial Ca^2+^ overload in MICU1 KO cells. 

Increasing the open duration of IP_3_R also has an opposite effect in WT (and MICU2 KO) and MICU1 KO mitochondria where we observe an increasingly positive and negative change in ATP, respectively. These results are in line with reports claiming that the human loss-of-function mutations in MICU1 results in early-onset neuromuscular weakness, impaired cognition, extrapyramidal motor disorder, and perinatal lethality [27,28,29,30,31] through mitochondrial Ca^2+^ overload and impaired bioenergetics [27]. The model also leads to two key observations that warrant further discussion. First, the Ca^2+^ influx in MICU1 KO mitoplasts results in significantly larger depolarization of IMM. This is in line with recent observations [18]. Specifically, HEK293 cells expressing two mutant MCU channels individually were exposed to repeated 5 to 7 μM boluses of cytosolic Ca^2+^ that were 150 s apart. While mitochondria expressing WT MCU rapidly took up Ca^2+^ in response to 9–12 successive boluses, those in cells expressing mutant MCU failed to take up Ca^2+^ after only two boluses and exhibited a complete loss of IMM potential. Interestingly, both mutants used in these experiments results in the loss of matrix Ca^2+^-mediated gatekeeping (inhibition) like MICU1 KO. 

Another key observation from the model is the significantly larger *V_NaCa,max_* in MICU1 KO mitochondria as compared to WT. This came out naturally from the fit as the Na^+^/Ca^2+^ exchangers have to compensate for larger Ca^2+^ influx in MICU1 KO mitochondria under resting conditions. Remarkably, a recent analysis of mitochondrial Ca^2+^-related genes in 25 publicly available microarray and RNA-Sequencing datasets revealed a significant upregulation of Na^+^/Ca^2+^ exchanger-encoding gene SLC8B1 in Alzheimer’s disease patients when compared to age-matched healthy individuals [51]. The authors attributed this to the counteracting effect in the diseased human brain to avoid mitochondrial Ca^2+^ overload. Along these lines, upregulating the expression level of SLC8B1 gene rescued mitochondrial Ca^2+^-induced pathology in 3xTg Alzheimer’s disease mouse model [73]. 

Given that kinetic data on the single channel function of MCU is slowly emerging [5,6,11,74], our model is formulated so that such future observations can be incorporated. For example, if time traces representing the gating state of MCU as a function of time is available, one can extract the transition rates between different states by fitting the following log-Likelihood function to the traces (see [34,75] for details): (13)$$\log\left(f\left({\mathrm{tc}}_{1},{\mathrm{to}}_{1},{\mathrm{tc}}_{2},{\mathrm{to}}_{2},\ldots \ldots {\mathrm{tc}}_{n},{\mathrm{to}}_{n}\right)\right)=\;\log\left(\begin{array}{c} \pi_{C}\exp\left(Q_{\mathrm{CC}}{\mathrm{tc}}_{1}\right)\;Q_{\mathrm{CO}}\exp\left(Q_{\mathrm{OO}}{\mathrm{to}}_{1}\right)\;Q_{\mathrm{OC}}\exp\left(Q_{\mathrm{CC}}{\mathrm{tc}}_{2}\right)\;Q_{\mathrm{CO}}\exp\left(Q_{\mathrm{OO}}{\mathrm{to}}_{2}\right)\ldots .\\ \exp\left(Q_{\mathrm{OO}}{\mathrm{to}}_{n}\right)u_{O} \end{array}\right),$$
where $\pi_{C}$ is a vector of the initial probabilities of all close states being occupied at equilibrium, *to_i_* and *tc_i_* are the *ith* open and close times in the time series, respectively, $u_{O}$ is a vector of all ones with length equal to the number of open states in the model, and *Q_OC_* is a (2 × 6) matrix of the transition rates from all open to all close states, etc. 

Alternatively, if the distributions of dwell-times in open or close states are available, one can extract the transition rates by fitting the following functions to such distributions, respectively [34,75]: (14)$$f_{O}\left(t_{O}\right)=\pi_{O}\exp(Q_{\mathrm{OO}}t_{O})Q_{\mathrm{OC}}u_{C},$$
(15)$$f_{C}\left(t_{C}\right)=\pi_{C}\exp(Q_{\mathrm{CC}}t_{C})Q_{\mathrm{CO}}u_{O}.$$
where $\pi_{O}$ is a vector of initial probabilities of all open states being occupied at equilibrium, $u_{C}$ is a vector of all ones with length equal to the number of close states in the model, *Q_CO_* is a (6 × 2) matrix of the transition rates from all close to all open states, and *Q_CC_* is a (6 × 6) matrix of the transition rates from all close to all close states etc. 

As a toy example, we used the double-exponential behavior of open dwell-time histogram observed in [11]. Specifically, in these experiments, MCU and the regulatory proteins were inserted into a planar bilayer membrane and electrophysiological recording of single channel activity was carried out. In one of these experiments (Figure S3B in [11]), the open dwell-time distribution could be fitted with double exponential with time constants τ_1_ = 12.06 ms and τ_2_ = 161.92 ms. We used these time constants to generate experimental dwell-time distribution (green bars in Figure 6A) and fit Equation (14) to it. The model fit to the data is shown by the dashed line in Figure 6A. The transition rates between different states from the fit are given in section “Extracting single channel gating parameters from dwell-time distributions” of the Appendix A. The rates obtained from the fit are then used to stochastically simulate the gating of single MCU channel at different [Ca^2+^]_C_ and [Ca^2+^]_M_ values using the method previously developed [35,76,77]. Consistent with the results at the whole-cell and single mitochondrion levels, the single channel activity of MCU increases progressively as we increase [Ca^2+^]_C_ (Figure 6B,C). Furthermore, the frequency of the channel visiting the open states in general, and O_44_ state in particular increases significantly as we increase [Ca^2+^]_M_ from 0.4 μM (Figure 6B) to 1 μM (Figure 6C). 

## 5. Conclusions 

Mitochondrial Ca^2+^ overload either due to MICU1 mutations [27,28,29,30,31], MCU upregulation [51], or impaired function of Na^+^/Ca^2+^ exchangers [73] plays a causal role in various diseases. For example, recent observations show that it not only contributes to the progression of disease but also precedes neuropathology in different animal models of Alzheimer’s disease [51,73]. Mitochondrial Ca^2+^ overload was shown to accelerate memory decline, amyloidosis, and tau pathology by promoting the production of reactive oxygen species, metabolic dysfunction, and upregulation of β secretase expression, and neurodegeneration [51,73]. Furthermore, preventing mitochondrial Ca^2+^ overload was proven sufficient to impede Alzheimer’s disease-associated pathology and memory loss [51,73]. A data-driven model such as ours will help understand the role of different mechanisms regulating mitochondrial Ca^2+^ uptake in Alzheimer’s disease and many other diseases where mitochondrial Ca^2+^ overload is believed to play a major role [78,79]. Our model not only incorporates several key observations about MCU function under WT, MICU1 KO, and MICU2 KO conditions at the whole-cell and single mitochondrion levels, but also leaves room for incorporating future observations and providing information about experimentally inaccessible parameters at the single MCU level such as the transition rates between different conducting states of the channel, etc.

## Figures and Tables

**Figure 1 cells-09-01520-f001:**
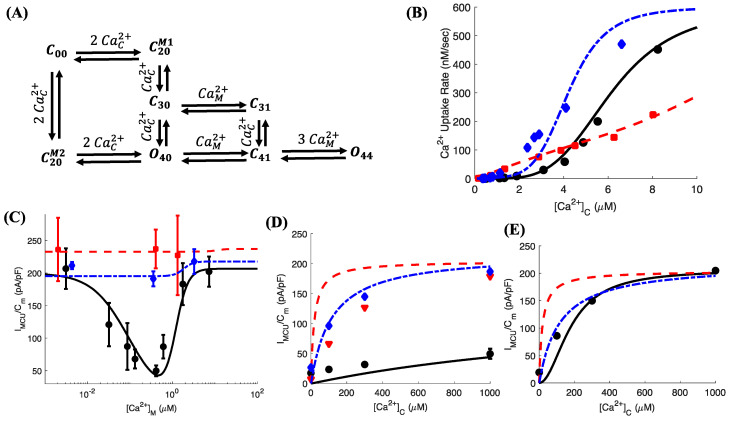
Scheme and fit to the experimental data of the MCU model. (**A**) The model consists of six close states (*C_XY_*) and two open (*O_XY_*) states. The subscripts *X* and *Y* represent the number of Ca^2+^ bound to domains of MCU on the cytosolic and matrix sides of the IMM. The superscripts M1 or M2 on the two close states with 2 Ca^2+^ bound on the cytosolic side indicate whether the ions are bound to the MICU1 EF hands or MICU2 EF hands. The channel has 2 Ca^2+^ bound to MICU1 and one Ca^2+^ bound to MICU2 on cytosolic side in state *C_3Y_*. (**B**) Ca^2+^ uptake rate as a function of [Ca^2+^]_C_ in WT (spheres, solid line), MICU1 KO (squares, dashed line), and MICU2 KO (diamonds, dashed-dotted line) cells. (**C**) MCU current density (I_MCU_/C_m_) as a function of [Ca^2+^]_M_ at [Ca^2+^]_C_ = 1 mM, (**D**) as a function of [Ca^2+^]_C_ at [Ca^2+^]_M_ = 0.4 μM, and (**E**) as a function of [Ca^2+^]_C_ at [Ca^2+^]_M_ = 0 μM in WT, MICU1 KO, and MICU2 KO mitoplasts. The lines and symbols use the same convention as in panel (**B**). Triangles in panel (**D**) represent MICU1 KD mitoplasts. Symbols and lines in panels (**B**–**E**) represent observed and theoretical values, respectively. Experimental data shown for comparison is from [12] (**B**) and [19] (**C**–**E**).

**Figure 2 cells-09-01520-f002:**
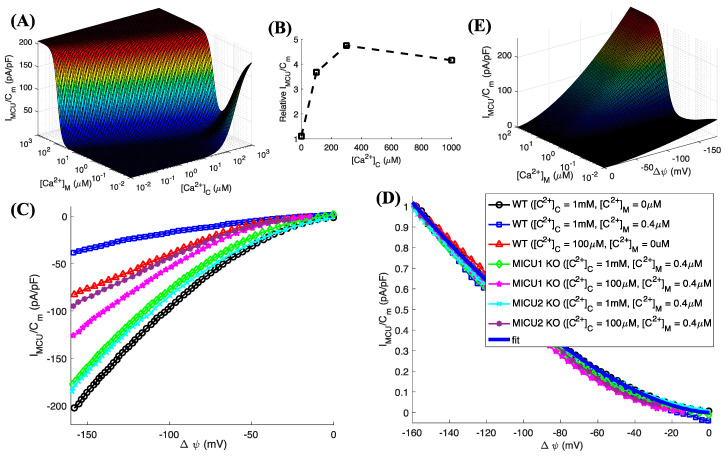
MCU current density as a function of [Ca^2+^]_C_, [Ca^2+^]_M_, and Δψ. (**A**) The inverted bell-shaped behavior of the MCU Ca^2+^ flux changes into a biphasic behavior with almost zero and optimal current density at low and high [Ca^2+^]_M_ respectively as we decrease [Ca^2+^]_C_; (**B**) the ratio of I_MCU_/C_m_ observed at [Ca^2+^]_M_ = 0 μM to that observed at [Ca^2+^]_M_ = 0.4 μM in WT mitoplasts at different [Ca^2+^]_C_ values. (**C**) raw and (**D**) normalized I_MCU_/C_m_ (normalized with respect to the minima (maximum Ca^2+^ influx) of the trace) at different [Ca^2+^]_C_ and [Ca^2+^]_M_ values as we vary Δψ in WT, MICU1 KO, and MICU2 KO mitoplasts. Thick solid line in (**D**) represents model fit; (**E**) I_MCU_/C_m_ given by the model as a function of [Ca^2+^]_M_ and Δψ at [Ca^2+^]_C_ = 100 μM. Experimental data shown for comparison is from [19] (**B**–**D**).

**Figure 3 cells-09-01520-f003:**
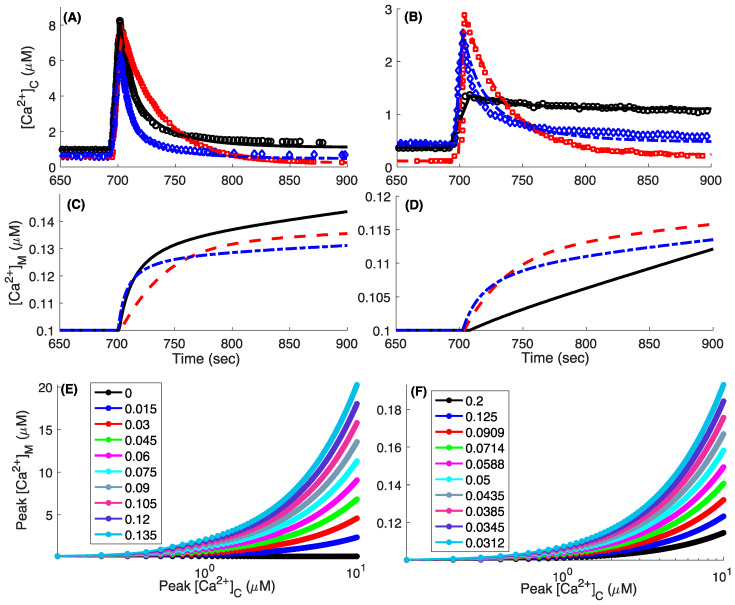
Model fits to the whole-cell cytosolic Ca^2+^ traces observed in the cell culture experiments and the effect of mitochondrial Ca^2+^ buffering capacity and relative size with respect to cytoplasm on free mitochondria Ca^2+^. (**A**) [Ca^2+^]_C_ in WT (spheres, solid line), MICU1 KO (squares, dashed line), MICU2 KO (diamonds, dashed-dotted line) cells. Symbols and lines represent experimental observations and model fits respectively. Cells were treated with 0.004% digitonin to permeabilize plasma membrane, 2 μM thapsigargin to block ER Ca^2+^ uptake, and 20 μM CGP37157 to inhibit mitochondrial Ca^2+^ efflux, added at t = 50, 100, and 400 s, respectively. After reaching steady state, a bolus of Ca^2+^ was added to the cytoplasm at ~700 s to raise [Ca^2+^]_C_ to above 7 μM. (**C**) [Ca^2+^]_M_ given by the model. Results in (**B**,**D**) are similar to (**A**,**C**) except that the bolus of Ca^2+^ added was smaller. Peak [Ca^2+^]_M_ 300 s after the Ca^2+^ bolus was added as a function of maximum [Ca^2+^]_C_ in WT cells as we change *f_m_* (**E**) and *δ* (**F**). The legends in (**E**,**F**) respectively represent the values of *f_m_* and *δ* used. In all simulations shown in Figure 3, a fixed mitochondrial membrane potential Δψ = −160 mV is used. Experimental data shown for comparison are from [12] (**A**,**B**).

**Figure 4 cells-09-01520-f004:**
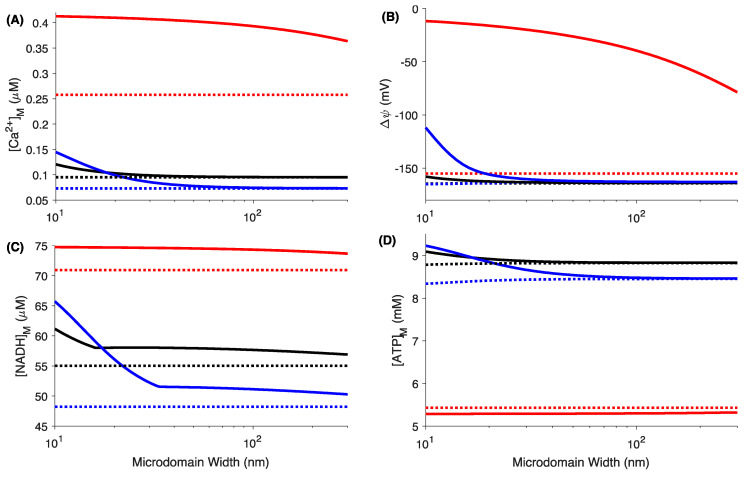
Changes in mitochondrial variables in response to the opening of a single IP_3_R channel at the single mitochondrion level. A single IP_3_R residing in the ER membrane is allowed to open for 100 ms starting at 100 s, and the maximum changes in different mitochondrial variables are recorded. The resting value (dotted lines) and the maximum value after the channel opens (solid lines) of [Ca^2+^]_M_ (**A**), Δψ (**B**), [NADH]_M_ (**C**), [ATP]_M_, and (**D**) in WT (black lines), MICU1 KO (red lines), and MICU2 KO cells (blue lines). Parameters used in the simulations are the same as given in the text. The value of *V_NaCa,max_* given by the model is 1.112, 1838.6, and 35.9675 μM/s.

**Figure 5 cells-09-01520-f005:**
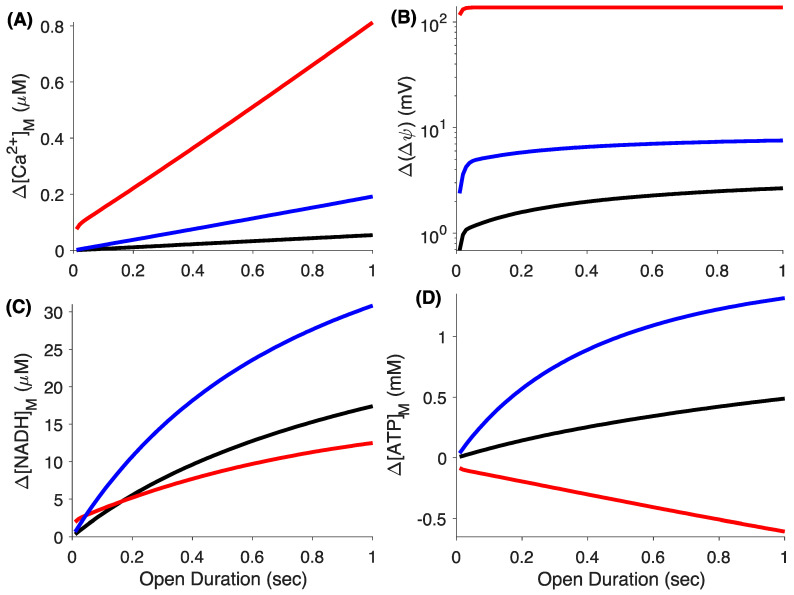
Relative changes in mitochondrial variables in response to the opening of a single IP_3_R channel at the single mitochondrion level. A single IP_3_R residing in the ER membrane is allowed to open for a given duration (*x*-axes) starting at 100 s and the optimal changes with respect to the resting values of different mitochondrial variables are recorded (see Appendix A) and shown along *y*-axes. The change in [Ca^2+^]_M_ (**A**), Δψ (**B**), [NADH]_M_ (**C**), and [ATP]_M_ (**D**). A 24 nm wide microdomain is considered in these simulations.

**Figure 6 cells-09-01520-f006:**
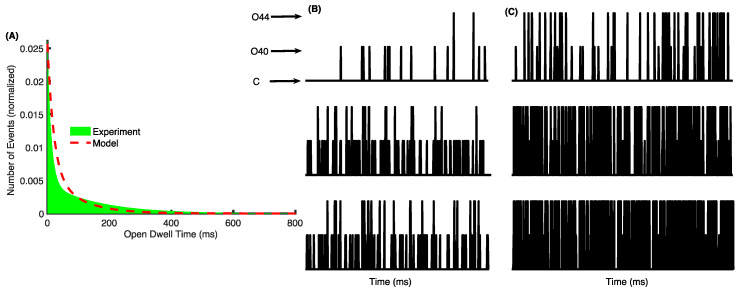
Extracting single channel kinetic parameters by fitting to the open dwell-time distribution. (**A**) Open dwell-time distribution generated using the time constants (τ_1_ = 12.06 ms and τ_2_ = 161.92 ms) observed in lipid bilayer experiments [11] (green bars) and model fit (dashed line); (**B**) time traces from the model using the transition rates obtained from the model at [Ca^2+^]_M_ = 0.4 μM and [Ca^2+^]_C_ = 1 μM (top panel), 10 μM (middle panel), and 100 μM (bottom panel); (**C**) same as in (**B**) but at [Ca^2+^]_M_ = 1 μM.

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
