# Peer review of "The Function of Mitochondrial Calcium Uniporter at the Whole-Cell and Single Mitochondrion Levels in WT, MICU1 KO, and MICU2 KO Cells"

_cells, 2020, doi:10.3390/cells9061520_

Round 1

Reviewer 1 Report

See in the file attached

Author Response

See the attached pdf.

Reviewer 2 Report

In their manuscript "The Function of Mitochondrial Calcium Uniporter at
3 the Whole-Cell and Single Mitochondrion Levels in
4 WT, MICU1 KO, and MICU2 KO Cells" Shah and Ullah investigated MCU function. They created a computational data-driven model that closely replicates the behavior of MCU and show how MICU1
or MICU2 KO affect mitochondrial function and show how it acts under condtions of different alcium load.

The model is comprehensiv and detailed and underscores the importance of the regulatory proteins. The data presentation is good and the model is relevant for mitochondrial function and in diseases where mitochondrial calcium overload plays a role.

Author Response

See attached pdf.

Reviewer 3 Report

In this study, Islamuddin and colleagues describe a data-driven model that reasonably predicts MCU behavior taking in consideration changes in several variables that contribute to mitochondrial calcium homeostasis including ΔΨ, NADH, ATP alongside genetic manipulation of MCU accessory MICU. While the theoretical background is robust and properly explained, the model faithfully recreates MCU behavior under different hypothetical scenarios as well as the recently proposed "flux sensor" by Foskett´s group. I only have minor concerns for this manuscript. 

1) Please include written permissions by Riley et al. as well as Vais et al. for the sake of good publishing practices. 

2) Please describe a bit in more detail the calcium "flux sensor".

3)  Explain why ΔΨ0= 91mV

4) Include the source of each experiment from the studies by Riley and Vais on each figure legend.

5) Fig. 2 A and 2E require a better color palette selection since printing in black and white makes the surface plots unreadable.  

6) The same applies for Fig. 3

Author Response

See attached pdf.
